# ATTRIBUTING DATA FOR SHARPNESS-AWARE MINIMIZATION

## ABSTRACT

Sharpness-aware Minimization (SAM) improves generalization in large-scale model training by linking loss landscape geometry to generalization. However, challenges such as mislabeled noisy data and privacy concerns have emerged as significant issues. Data attribution, which identifies the contributions of specific training samples, offers a promising solution. However, directly rendering existing data influence evaluation tools such as influence functions (IF) to SAM will be inapplicable or inaccurate as SAM utilizes an inner loop to find model perturbations that maximize loss, which the outer loop then minimizes, resulting in a doubled computational structure. Additionally, this bilevel structure complicates the modeling of data influence on the parameters. In this paper, based on the IF, we develop two innovative data valuation methods for SAM, each offering unique benefits in different scenarios: the Hessian-based IF and the Gradient Trajectory-based IF. The first one provides a comprehensive estimation of data influence using a closed-form measure that relies only on the trained model weights. In contrast, the other IF for SAM utilizes gradient trajectory information during training for more accurate and efficient data assessment. Extensive experiments demonstrate their effectiveness in data evaluation and parameter tuning, with applications in identifying mislabeled data, model editing, and enhancing interpretability.

## 1 INTRODUCTION

Over the past decade, deep neural networks have advanced significantly due to increased model parameter sizes and improved training algorithms that enhance generalization. However, larger models often memorize training data, leading to overfitting and poor generalization. In order to address this issue, considerable effort has been invested in the development of a range of strategies, including regularization techniques (Wu et al., 2021; Yoshida & Miyato, 2017), adversarial training (Mądry et al., 2017; Shafahi et al., 2019), model uncertainty (Gal & Ghahramani, 2016; Blundell et al., 2015), and neural architecture search (Zoph, 2016). Recent work has observed that sharp local minima in the loss landscape can significantly impair the generalization performance of deep networks (Keskar et al., 2016; Hochreiter & Schmidhuber, 1994; Neyshabur et al., 2017). To make loss landscape flatter to improve models' generalization ability, Foret et al. (2020) introduced a general framework called Sharpness-aware Minimization (SAM). SAM improves generalization by penalizing sharp minima and encouraging convergence to flatter regions. Intuitively, SAM is a bilevel optimization problem, where the inner level seeks weight perturbations that can lead to the maximum loss, which is a measure of local sharpness. On the outer level, the model is trained to minimize both loss and local sharpness simultaneously. Thus, it can also be formulated as a minimax optimization problem. SAM has achieved state-of-the-art results across various tasks (Foret et al., 2020; Chen et al., 2021; Liu et al., 2022).

While SAM has been applied in various real-world applications (Du et al., 2021; Andriushchenko et al., 2023; Qu et al., 2022; Bahri et al., 2021), the presence of noisy data in the training set, including mislabeled or poisoned data, has become a significant concern. A critical approach to tackling this issue involves identifying the contributions of training samples in SAM-trained models by assessing their impact on model performance, a process referred to as (training data) attribution. Data attribution plays a critical role in tracing model outputs back to significant training examples, thereby providing insights into how individual data points influence model performance.

Data attribution has been used in various tasks to enhance model transparency and understanding of how training data influences model behavior (Ribeiro et al., 2016; Lundberg, 2017). Generally, data attribution methods (Jia et al., 2019; Ghorbani & Zou, 2019; Yoon et al., 2020; Han et al., 2020) assign higher contribution scores to training instances that significantly improve model performance when included, which can be divided into two types. The first one, such as Shapley Value (Winter, 2002), is based on sampling (Lundberg, 2017; Kwon & Zou, 2022), which requires multiple retraining with different data subsets. This is computationally expensive and impractical for large models. To address this challenge, the second approach—namely, influence function-based methods (Koh & Liang, 2017; Feldman & Zhang, 2020)—estimates data contributions using gradient information, thereby facilitating accurate assessments without the need for retraining. Recent advancements have led to innovative estimators utilizing the training gradient trajectory (Pruthi et al., 2020; Schioppa et al., 2024), which estimate the influence of training data on model predictions by tracing gradient descent, providing better insights into and improvements of training examples in deep learning models. These methods do not depend on the convexity assumption of the loss function, nor do they require computationally intensive Hessian inversions. Instead, they perform calculations based on the training gradient and have achieved good results in many tasks (Liu & Yang, 2024).

However, it is essential to note that influence functions were initially designed for M-estimators (Huber, 1981) (i.e., minimizing the summation of loss on all data), which limits their direct applicability to the bilevel structure in SAM. Specifically, the influence of data on perturbations, and consequently on model parameters, is indirect manner and has not been considered in classical influence functions. Due to the coupled outer-inner optimization process involved in SAM, any changes to the model parameters will necessitate further updates to the inner perturbations and consequently alter the outer model parameters. Furthermore, in the actual training of SAM, the inner perturbations are not derived through the optimization process but are instead obtained via approximation. To further accelerate computations, the outer gradient descent is also approximated, resulting in minor deviations in the inner perturbations from the original SAM model. These complicate our ability to evaluate the influence of data in SAM accurately.

In this paper, we propose two data valuation methods based on influence functions for SAM, focusing on two distinct scenarios. Firstly, we derive a closed form of the Hessian-based Influence Function (SAM-HIF) from the mathematical formulation of SAM through theoretical derivation. SAM-HIF fully considers the impact of data on the model's local sharpness. In the absence of gradient trajectory information, SAM-HIF serves as a comprehensive estimation method. However, when gradient trajectory information during training is accessible, we can focus on the specific training algorithm to derive a more accurate evaluation method. Inspired by Pruthi et al. (2020), we propose the Gradient Trajectory-based Influence Function (SAM-GIF). This method utilizes checkpoints to evaluate the influence of data within SAM, leading to accurate data assessment while reducing implementation and computational costs. SAM-GIF proves to be highly effective due to its simplicity, high scalability, and accuracy. To sum up, our contributions are listed as follows:

- We present data evaluation methods using influence functions (IF) for SAM and its training algorithms. First, we derive SAM-HIF, which allows for precise modeling of data influence in SAM by effectively capturing influences on both model parameters and model perturbations. Second, we present SAM-GIF, which showcases exceptional precision and scalability when training gradient trajectories are accessible. We also discuss some downstream applications and strategies to accelerate the computation of our influence functions.

- Our method has potential applications in several real-world scenarios, including the automatic identification and removal of potentially harmful data points, SAM model editing, and enhancing the interpretability of SAM-trained models. We validate the effectiveness and efficiency of SAM-HIF and SAM-GIF through comprehensive experimental evaluations. Experimental results indicate that our framework performs well in data evaluation and the tuning of SAM-trained models.

## 2 RELATED WORK

**Sharpness Aware Minimization.** The concept of flat minima and its link to reducing overfitting and enhancing model generalization was first explored by Hochreiter & Schmidhuber (1997), establishing a foundation for understanding why models that converge to flatter regions tend to generalize

better. Additionally, Keskar et al. (2016) experimentally examined the relationship between batch size, sharp minima, and generalization. Their findings provided empirical support for SAM's theoretical foundations and highlighted the importance of sharpness in optimization and model training. Overall, sharp local minima can significantly influence the generalization capabilities of deep networks (Chaudhari et al., 2019; Izmailov et al., 2018). Building on these insights, one of the foundational works on SAM is by Foret et al. (2020), who proposed a bi-level optimization framework that seeks perturbations around model parameters to maximize loss and guide the model towards flatter minima. Recent studies have further explored the behavior SAM. For example, Andriushchenko & Flammarion (2022) investigated the geometric properties of loss landscapes, demonstrating that SAM's perturbation-based approach offers insights into model behavior. Their findings indicate that SAM not only improves generalization but also enhances understanding of models' sensitivity to input perturbations. Zhao et al. (2022) assert that SAM (Foret et al., 2020) is effectively equivalent to applying a gradient norm regularization through the approximation of the Hessian matrix. Kwon et al. (2021) introduce adaptive SAM, which can dynamically adjust the maximization region based on the weight scale. To minimize computational costs associated with SAM, Du et al. (2021) introduced Efficient SAM, which randomly computes perturbations.

**Influence Function.** The influence function(IF), initially developed in robust statistics (Cook, 2000; Cook & Weisberg, 1980), has become essential in machine learning since its introduction by Koh & Liang (2017). IFs have been used in various fields, including interpreting model outputs, reducing model bias (Wang et al., 2019), and facilitating machine unlearning (Liu et al., 2024; Golatkar et al., 2020; 2021). Its versatility spans various fields, including natural language processing (Han et al., 2020) and image classification (Basu et al., 2021), while also addressing biases in classification models (Wang et al., 2019), word embeddings (Brunet et al., 2019), and model fine-tuning (Chen et al., 2020). A recent innovative influence function that leverages the training gradient trajectory has been proposed and investigated (Pruthi et al., 2020; Schioppa et al., 2024), and has been successfully applied in instructional fine-tuning (Xia et al., 2024). Despite the numerous studies related to influence functions, we are the first to use this concept for the data evaluation of models trained for SAM. This provides us with a completely new perspective on understanding the training process of SAM.

## 3 PRELIMINARIES

**Sharpness-Aware Minimization (SAM).** SAM presents a novel approach to optimizing machine learning models with the primary objective of enhancing generalization by mitigating the sharpness of the loss landscape. Given a dataset $S = \{(x_i, y_i)\}_{i=1}^n$, we define the training loss associated with model parameters $\omega$ and a perturbation $\epsilon$ as:

$$L_S(\omega + \epsilon) = \frac{1}{n} \sum_{i=1}^n \ell(x_i, y_i; \omega + \epsilon) \triangleq \sum_{i=1}^n L_S^i(\omega + \epsilon). \tag{1}$$

Then SAM optimization framework seeks parameters that lie in neighborhoods having uniformly low loss by the following procedure. Firstly, seeking the model perturbation maximizing the loss $\hat{\epsilon}(\omega) = \arg\max_{||\epsilon||_p \leq \rho} L_S(\omega + \epsilon)$. Then minimize the uniform loss:

$$\omega^* = \arg\min_\omega L_S(\omega + \hat{\epsilon}(\omega)) + \frac{\lambda}{2} \cdot ||\omega||_2^2. \tag{2}$$

where $\lambda$ is the regularization parameter. Here, SAM takes model loss sharpness into consideration via including the perturbation process. The loss function for SAM can be formed into one equation as follows:

$$L_S^{SAM}(\omega) = L_S(\omega + \hat{\epsilon}(\omega)) + \frac{\lambda}{2} \cdot ||\omega||_2^2$$

$$= \max_{||\epsilon||_p \leq \rho} L_S(\omega + \epsilon) + \frac{\lambda}{2} \cdot ||\omega||_2^2.$$

In essence, SAM seeks to minimize the worst-case loss by enforcing a level of robustness against perturbations in the parameter space, thereby promoting smoother loss landscapes. Besides, the gradient of $L_S^{SAM}$ is calculated by

$$\nabla L_S^{SAM}(\omega) = \nabla L_S(\omega + \hat{\epsilon}(\omega)) + \frac{d\hat{\epsilon}(\omega)}{d\omega} \nabla L_S(\omega + \hat{\epsilon}(\omega)). \tag{3}$$

**Influence Function.** The influence function (Huber, 1981) quantifies how an estimator relies on the value of each individual point in the sample. Consider a neural network $\hat{\theta} = \arg\min_\theta L(\theta, D) = \sum_{i=1}^n \ell(z_i; \theta)$ with loss function $\ell$ and dataset $D = \{z_i\}_{i=1}^n$. When an individual data point $z_m$ is removed from the training set, the retrained optimal retrained model is denoted as $\hat{\theta}_{-z_m}$. The influence function method provides an efficient way to approximate $\hat{\theta}_{-z_m}$ without the need of retraining. By increasing the weight of the $z_m$ loss term by $\delta$, define a series of $\delta$-parameterized optimal models by $\hat{\theta}_{-z_m,\delta} = \arg\min_\theta [L(\theta, D) + \delta\, \ell(z_m; \theta)]$. Consider the term $\nabla L(\hat{\theta}_{-z_m,\delta}, D) + \delta \cdot \nabla \ell(z_m; \hat{\theta}_{-z_m,\delta}) = 0$, we perform a Taylor expansion at $\hat{\theta}$ and incorporate the optimal gradient condition at $\hat{\theta}_{-z_m}$ and $\hat{\theta}$:

$$\sum_{i=1}^n \nabla \ell(z_i; \hat{\theta}) + \delta \cdot \nabla \ell(z_m; \hat{\theta}) + H_{\hat{\theta}} \cdot \left( \hat{\theta}_{-z_m,\delta} - \hat{\theta} \right) \approx 0$$

where $H_{\hat{\theta}} = \sum_{i=1}^n \nabla_{\hat{\theta}}^2 \ell(z_i; \hat{\theta})$ is the Hessian matrix. Consequently, the Influence Function is defined as the derivative of the change in parameters of the retrained model due to perturbation with respect to the perturbation:

$$\mathrm{IF}(z_m) = \left. \frac{\mathrm{d}\hat{\theta}_{-z_m,\delta} - \hat{\theta}}{\mathrm{d}\delta} \right|_{\delta=0} \approx -H_{\hat{\theta}}^{-1} \cdot \nabla \ell(z_m; \hat{\theta}).$$

When setting $\delta = -1$, this results in the complete removal of $z_m$ from the retraining process. Then, $\hat{\theta}_{-z_m}$ can be approximated by a linear approximation formula as $\hat{\theta} - \mathrm{IF}(z_m)$. Additionally, for a differentiable evaluation function, such as one used to calculate the total model loss over a test set, the change resulting from up-weighting $\epsilon$ to $z_m$ in the evaluation results can be approximated as $-\nabla f(\hat{\theta}) \cdot \mathrm{IF}(z_m)$.

# 4 EVALUATING DATA ATTRIBUTION IN SAM

To evaluate the influence of an individual data point for SAM, we utilize assessing model differences after leave-one-out (LOO) retraining.

## 4.1 DATA ATTRIBUTION IN SAM VIA HESSIAN IF

To evaluate the influence of an individual data point for the model trained via SAM, we first provide an estimation of the leave-one-out (LOO) retrained model. Then we can quantify the parameter-level influence of the excluded data point by analyzing the differences in model parameters before and after the retraining process. Due to the unique architecture of SAM, the form of the corresponding loss function differs from that of traditional IF. This is because SAM includes an additional step to find parameter perturbations that maximize the loss, which will also change as a result of the LOO retraining process. We provide details in the following.

**Definition 4.1** (LOO Retrained Parameter). For $(x_k, y_k)$ to be evaluated, let $\hat{\epsilon}_k(\omega)$ be the perturbation maximizing the loss after removing $(x_k, y_k)$, defined as $\hat{\epsilon}_k(\omega) = \arg\max_{||\epsilon||_p \leq \rho} \sum_{i \neq k}^n L_S^i(\omega + \epsilon)$. The retrained model is defined as $\omega_k = \arg\min_\omega \sum_{i \neq k} L_S^i(\omega + \hat{\epsilon}_k(\omega)) + \frac{\lambda}{2} \cdot ||\omega||_2^2$.

From the preceding discussion, we denote the influence of the data point $(x_k, y_k)$ on the SAM-trained model as $\omega_k - \omega^*$. To avoid retraining, we utilize the influence function method to approximate this difference. We begin by transforming the optimality condition of (2) into a simpler form via Danskin's Theorem (Danskin, 2012).

**Lemma 4.2.** *The optimal solution $w^*$ of SAM, as represented in (2), satisfies $\nabla L_S(\omega^* + \hat{\epsilon}(\omega^*)) + \frac{\mathrm{d}\hat{\epsilon}(\omega^*)}{\mathrm{d}\omega^*} \nabla L_S(\omega^* + \hat{\epsilon}(\omega^*)) + \lambda\omega^* = 0$, which is equivalent to $\nabla L_S(\omega^* + \hat{\epsilon}(\omega^*)) = 0$.*

Follow the idea of influence function, we up-weigh the $k$-th term in the loss function by a factor of $\delta$. This adjustment allows us to derive a series of model parameters $\omega_{k,\delta}$ obtained by training the models via SAM:

$$\omega_{k,\delta} = \arg\min L_{\mathrm{Total},\delta}(\omega) \triangleq L_{S,\delta}(\omega) + \frac{\lambda}{2} \cdot ||\omega||_2^2 = L_S(\omega + \hat{\epsilon}_\delta(\omega)) + \delta \cdot L_S^k(\omega + \hat{\epsilon}_\delta(\omega)) + \frac{\lambda}{2} \cdot ||\omega||_2^2$$

The $\delta$-related term in the loss function captures the influence of $(x_k, y_k)$ on $\omega$ (where $\omega_{k,-1} = \omega_k$). Moreover, $\delta$ also affects the worst-case perturbation through the modified objective for $\epsilon$: $\hat{\epsilon}_{k,\delta}(\omega) = \arg\max_{||\epsilon||_p \leq \rho} \left[ \sum_{i=1}^n L_S^i(\omega + \epsilon) + \delta \cdot L_S^k(\omega + \epsilon) \right]$. This modification in the objective function, in turn, indirectly influences the optimal parameters learned through SAM. Starting from a simplified case, we neglect the indirect influence of the perturbation and introduce the simplified version of SAM-IF.

**Theorem 4.3.** *For the $k$-th data point $(x_k, y_k)$ and a SAM-trained model $\omega^*$, $\hat{\epsilon}(\omega^*)$ represents the model weight perturbation that results in the largest loss, the corresponding SAM-IF is given by: SAM-IF$(x_k, y_k) = -H_\omega^{-1} \cdot \nabla L_S^k(\omega^* + \hat{\epsilon}(\omega^*))$ where $H_\omega$ is defined as $H_\omega = \nabla^2 L_S(\omega^* + \hat{\epsilon}(\omega^*)) + \lambda$. $\omega_k$ can be approximated by $\omega_k \approx \omega^* - $ SAM-IF$(x_k, y_k)$.*

We can see that in SAM-IF we still use $\hat{\epsilon}(\omega^*)$ in both Hessian matrix and gradient, while the optimal perturbation should be $\hat{\epsilon}_{k,-1}(\omega_k)$. However, neglecting the term related to the perturbation can lead to an incomplete and inaccurate evaluation of data influence. In fact, it implies that while the influence of this data point on the model's loss function is considered, the impact of removing this data point on the computation of model sharpness is not taken into account. This will lead to an incomplete evaluation of data influence. To take this into consideration, $\hat{\epsilon}_{k,\delta}(\omega_{k,\delta}) - \hat{\epsilon}(\omega^*)$ also need to be estimated, which is given in the following lemma.

**Lemma 4.4.** $\hat{\epsilon}_{k,\delta}(\omega_{k,\delta}) - \hat{\epsilon}(\omega^*) \approx \hat{\epsilon}_{k,\delta}(\omega^*) - \hat{\epsilon}(\omega^*) + \left. \frac{\mathrm{d}\hat{\epsilon}_{k,\delta}(\omega)}{\mathrm{d}\omega} \right|_{\omega=\omega^*} \cdot (\omega_{k,\delta} - \omega^*)$.

When $\epsilon \to 0$, $\hat{\epsilon}_{k,\delta}(\omega^*) - \hat{\epsilon}(\omega^*) \to 0$, then $\hat{\epsilon}_{k,\delta}(\omega_{k,\delta}) - \hat{\epsilon}(\omega^*)$ can be bounded by $\omega_{k,\delta} - \omega^*$, which is especially useful for the following derivation.

**Theorem 4.5.** *For the $k$-th data point $(x_k, y_k)$ and a SAM-trained model $\omega^*$, the SAM-HIF is given by $-\left( H_\omega + H_\omega \frac{\mathrm{d}\hat{\epsilon}(\omega^*)}{\mathrm{d}\omega} \right)^{-1} \nabla L_S^k(\omega^* + \hat{\epsilon}(\omega^*))$. Then $\omega_k \approx \omega^* - $ SAM-HIF$(x_k, y_k)$.*

Note that in general, there is no closed-form solution for the term $\frac{\mathrm{d}\hat{\epsilon}(\omega^*)}{\mathrm{d}\omega}$, to address the issue, we can use the method in (Foret et al., 2020) to approximate $\hat{\epsilon}$:

$$\hat{\epsilon}(\omega) = \rho \cdot \text{sign}(\nabla_w L_S(\omega, 0)) \cdot \frac{|\nabla_w L_S(\omega, 0)|^{q-1}}{(\|\nabla_w L_S(\omega, 0)\|_q^q)^{1/p}}, \tag{4}$$

where $p$ satisfies $\frac{1}{p} + \frac{1}{q} = 1$. Assuming the sign of the gradient $\nabla_w L_S(\omega, 0)$ remains unchanged when one data point is removed, $\frac{\mathrm{d}\hat{\epsilon}(\omega^*)}{\mathrm{d}\omega^*}$ can be calculated easily.

## 4.2 IMPROVING DATA VALUATION VIA GRADIENT TRAJECTORY

In Theorem 4.5, we have to know the exact worst perturbation $\hat{\epsilon}(\omega)$. However, according to (Foret et al., 2020), to accelerate the training, the training algorithm for SAM uses the following closed-form estimation of $\hat{\epsilon}(\omega^*)$ in (4) to replace the procedure of finding the model perturbation that results in the largest loss. Besides, for the outer-level gradient descent, the second term in (3) is intentionally dropped during training for simplification, and the final gradient approximation they use is actually $\nabla L_S(w + \hat{\epsilon}(w))$. See Algorithm 1 for details. Therefore, the SAM-HIF we proposed based on the minimization condition represents an idealized case, which sometimes proves insufficiently reliable in practical applications where we cannot find the optimal minimizer. To bridge the gap, we will next focus on the SAM model trained via the canonical training algorithm in (Foret et al., 2020), where we have the training trajectory stored in checkpoints. To measure the influence of the $k$-th data in the training dataset, similar to HIF, we firstly define a loss function with the $k$-th term up-weighted by $\delta$ as:

$$L_S(\omega; \delta) = L_S(w) + \delta \cdot L_S^k(\omega). \tag{5}$$

Consider the model trained based on loss function (5) using a gradient descent optimizer in (Foret et al., 2020). At training step $t$, model parameters $\omega_{t,\delta}$ are updated as

$$\omega_{t-1,\delta} - \eta_{t-1} \cdot \nabla_\omega L_S(\omega_{t-1} + \hat{\epsilon}(\omega_{t-1})) - \delta \cdot \eta_{t-1} \nabla_\omega L_S^k(\omega_{t-1,\delta} + \hat{\epsilon}(\omega_{t-1,\delta})),$$

where $\eta_{t-1}$ is the learning rate at step $t-1$, and $\hat{\epsilon}(\omega_{t-1}, \delta)$ is defined in (4). Then, the model trained after $T$ steps becomes $\omega_{T,\delta} = \omega_0 - \sum_{t=0}^{T-1} \eta_t \cdot \nabla_\omega L_S(\omega_{t,\delta} + \hat{\epsilon}(\omega_{t,\delta}); \delta)$. Based on the

above discussion, we derive the Gradient-based Influence Function (SAM-GIF) by calculating the derivative of the up-weighted retrained parameter $\omega_{T,\delta}$ for the $k$-th data point with respect to $\delta$.

$$\frac{\mathrm{d}\omega_{T,\delta}}{\mathrm{d}\delta}\bigg|_{\delta=0} = -\sum_{t=0}^{T-1} \eta_t \cdot \frac{\mathrm{d}\nabla_\omega L_S\left(\omega_{t,\delta} + \hat{\epsilon}(\omega_{t,\delta}); \delta\right)}{\mathrm{d}\delta}\bigg|_{\delta=0} - \sum_{t=0}^{T-1} \eta_t \cdot H_{t,0} \cdot \frac{\mathrm{d}(\omega_{t,\delta} + \hat{\epsilon}(\omega_{t,\delta}))}{\mathrm{d}\delta}\bigg|_{\delta=0},$$

where $H_{t,0}$ is the Hessian matrix of $L_S(\omega_t)$. To reduce the calculation complexity, we omit the second Hessian term and thus have the following approximation:SAM-GIF$_{\mathrm{GD}}(x_k, y_k) = -\sum_{t=0}^{T-1} \eta_t \cdot \nabla_\omega L_S^k(\omega_t + \hat{\epsilon}(\omega_t))$. This definition is consistent with (Pruthi et al., 2020). However, compared with gradient descent, SAM is more often carried out under the framework of SGD optimizer. Under this setting, it is possible that $(x_k, y_k)$ may not be used in some updating steps. Therefore, we utilize $B_{k,t}$ to indicate whether $(x_k, y_k)$ is used in the $t$-th gradient descent step. Finally, we have the following result.

**Theorem 4.6.** *Assume the model is trained by SGD with the worst perturbation $\hat{\epsilon}(\cdot)$ in each iteration is calculated via (4). For the data $(x_k, y_k)$, it Gradient trajectory-based IF (SAM-GIF) is given by SAM-GIF$_{SGD}(x_k, y_k) = \sum_{t=0}^{T-1} \eta_t B_{k,t} \nabla_\omega L_S^k(\omega_t + \hat{\epsilon}(\omega_t))$. Then, the LOO retrained model $\omega_k$ under SAM can be estimated by $\omega_k \approx \omega^* - $ SAM-GIF$_{SGD}(x_k, y_k)$.*

### 4.3 Computation Acceleration

The SAM-HIF outlined in Section 4.1 requires calculations of the inverse Hessian-vector product (iHVP). To enhance the scalability of SAM-HIF, we will introduce one efficient acceleration technique to expedite the computation of iHVP.

**Neumann Series Approximation Method.** The calculation in Proposition 4.3 and Theorem 4.5 is expressed as $-H^{-1} \cdot G$, where $H$ denotes the Hessian and $G$ is the gradient. Then, with the help of the Neumann series: $H^{-1} \cdot G = (I - (I - H))^{-1} \cdot G = G + \sum_{j=1}^{+\infty} (I - H)^j \cdot G$. By truncating this series at order $J$, we derive approximation as $H^{-1} \cdot G \approx G + (I - H) \cdot G + \cdots (I - H)^J \cdot G$. It is important to clarify that we did not specifically focus on accelerating the inversion of the Hessian matrix; instead, we optimized the iHVP directly. This approach eliminates the need to store Hessian matrix, which significantly reduces the memory requirements of our method.

### 4.4 Practical Applications

This section presents downstream tasks leveraging SAM-HIF and SAM-GIF in diverse scenarios.

**Model Editing.** We can use IFs to update the model under the removal of certain data. Specifically, the model after removing the $k$-th data point can be estimated as $\omega^* - $ SAM-HIF$(x_k, y_k; \omega^*)$. Additionally, if the training gradient trajectory is available, we can obtain an efficient and accurate estimate as $\omega^* - $ SAM-GIF$(x_k, y_k; \omega^*)$.

**Data Evaluation.** By appropriately selecting the model evaluation function, we can define influence scores (ISs) for individual data points. These scores can then be applied to data selection to enhance SAM performance.

**Definition 4.7. (Evaluation Function)** Given a validation dataset defined as $D_{val} = \{(x_t, y_t)\}_{t=1}^n$. the SAM learned parameter $\omega^*$ performance on the test task is defined as $\sum_{(x,y)\in D_{val}} \ell(x, y; \omega^*)$.

Thus, we propose an influence evaluation method.

**Theorem 4.8.** *Given a validation dataset defined as $D_{val} = \{(x_t, y_t)\}_{t=1}^n$. Denote the SAM-retrained model after the removal of $(x_k, y_k)$ as $\omega_{-k}^*$, then*

$$\sum_{(x,y)\in D_{val}} \ell(x, y; \omega^*) - \sum_{(x,y)\in D_{val}} \ell(x, y; \omega_{-k}^*) \approx \sum_{(x,y)\in D_{val}} \nabla\ell(x, y; \omega^*) \cdot IF(x, y) \triangleq IS(x, y),$$

*where IF can be SAM-HIF or SAM-GIF. We define the right hand of the above equation as the Influence Score (IS).*

A positive IS indicates that removing the data point will deteriorate the model's performance on the test dataset. Thus, this data point is valuable for model performance. Assigning an IS to each training data point allows us to identify useful and harmful data for the model's performance. Detailed theoretical derivations are provided in the Appendix.

Table 1: Performance Comparison on CIFAR-10, CIFAR-100, and MINI-Imagenet.

| Method | CIFAR-10 | | CIFAR-100 | | MINI-Imagenet | |
| --- | --- | --- | --- | --- | --- | --- |
| | Accuracy | RT (s) | Accuracy | RT (s) | Accuracy | RT (s) |
| Retrain | 0.9500(61) | 3516.74(845) | 0.7890(311) | 3244.25(218) | 0.6835(460) | 682.56(791) |
| SAM-HIF(Fast) | 0.9323(110) | 11.07(341) | 0.7208(142) | 13.07(416) | 0.6478(125) | 11.32(243) |
| SAM-HIF | 0.9443(121) | 41.50(321) | 0.7213(239) | 42.41(324) | 0.6516(213) | 39.21(309) |
| SAM-GIF | 0.9497(142) | 4.89(142) | 0.7227(469) | 6.89(231) | 0.6446(122) | 5.89(112) |

## 5 EXPERIMENTS

In this section, we demonstrate our main experimental results on utility, efficiency, effectiveness, and the abilities to identify harmful data and enhance interpretability. Details and additional results are in Appendix due to space limit.

### 5.1 EXPERIMENTAL SETTINGS

**Dataset.** We evaluated our algorithm on seven datasets: CIFAR-10, CIFAR-100 (Alex, 2009), Mini-ImageNet (Deng et al., 2009), MNIST (LeCun et al., 1998), HAM10000 (Tschandl et al., 2018), CUB (Wah et al., 2011), and Food101 (Bossard et al., 2014). CIFAR-10/100 are benchmarks for small-scale image classification, MiniImageNet is widely used for few-shot learning, and MNIST comprises handwritten digits. HAM10000, CUB, and Food101 assess generalization on skin lesions, bird species, and food recognition, respectively.

**Baselines.** We use retraining as the ground truth: removing a data point and retraining the model with SAM. Baselines include TRAK (Park et al., 2023), a projection-based kernel approximation for large-scale data attribution, and IF-EKFAC (Grosse et al., 2023), which leverages EKFAC to efficiently approximate Hessian-based influence functions.

When the training trajectory is unavailable, we use SAM-HIF and SAM-HIF (fast), implementing Theorems 4.3 and 4.5, respectively, and accelerate both with the Neumann Series approximation (Section 4.3). When the trajectory is available, we use SAM-GIF, implementing Theorem 4.6. Detailed algorithms are provided in the Appendix.

**Evaluation Metric.** We used two primary evaluation metrics to assess our models: accuracy and runtime (RT). Accuracy evaluates the model's performance by measuring the proportion of correctly classified instances out of the total instances. Runtime(RT), measured in seconds, assesses the time required for each method to update the model.

**Implementation Details.** We conducted experiments using the Nvidia RTX 4090-24G GPU. For all experiments, we selected the WideResNets architecture as the backbone network for the classification task. For utility evaluation, we randomly selected samples at different proportions from four datasets. For valuable (or harmful) samples, we removed 0-10 % of the data identified as valuable (or harmful) by the algorithm. Additionally, the removal process was repeated five times using different random seeds to obtain experimental results.

### 5.2 EVALUATION OF UTILITY AND EDITING EFFICIENCY

We first demonstrate the results on accuracy and time consumption of three algorithms, SAM-HIF (fast), SAM-HIF, and SAM-GIF, against the retrain method. Our experimental results are presented in Table 1. It is evident that our three proposed algorithms significantly improve computational efficiency without sacrificing accuracy. On the CIFAR-10 dataset, the time cost of retraining reached 3516.74, while our methods notably enhance computational efficiency, with SAM-HIF (fast), SAM-HIF, and SAM-GIF reducing the time to 11.0698, 41.4960, and 4.894 seconds, respectively. The accuracy differences between SAM-HIF (fast), SAM-HIF, and SAM-GIF compared to retrain are 0.0177, 0.0057, and 0.0003, respectively. These findings suggest that SAM-HIF (fast), SAM-HIF, and SAM-GIF can save substantial computational time required for retraining while achieving comparable accuracy.

### 5.3 HELPFUL DATA REMOVAL: TASK-RELATED IS ACCURACY

Furthermore, we observe that by avoiding the computation of the Hessian matrix, SAM-GIF requires less time than both SAM-HIF (Fast) and SAM-HIF. Besides, SAM-GIF not only surpasses SAM-HIF and SAM-HIF (Fast) in speed but also achieves accuracy comparable to retraining. This corroborates our previous discussion: compared to the SAM model, gradient trajectories can more accurately reflect the training of the SAM model, leading to a more precise estimation result. By avoiding the computation of the Hessian matrix, SAM-GIF demonstrates better

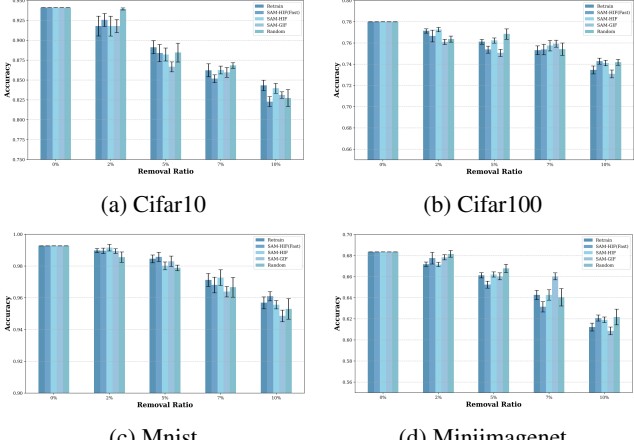

(a) Cifar10      (b) Cifar100

(c) Mnist      (d) Miniimagenet

Figure 1: Helpful Data Removal: Task-Related IS Accuracy

time efficiency than other methods while also showing improved accuracy on CIFAR-10. These findings indicate that utilizing gradient information effectively enhances both training efficiency and predictive accuracy. Moreover, we can see SAM-HIF achieves higher accuracy than SAM-HIF (Fast) by incorporating perturbation-related components into the Hessian matrix; however, this comes at the cost of a 2-3× increase in runtime. Mathematically, as we mentioned in Theorem 4.5, SAM-HIF further considers the change of perturbations due to the data removal. This can make it evaluate data influence more accurately but with additional computation time.

## 5.4 EVALUATION OF EFFECTIVENESS

Here, we aim to show that our proposed influence functions can be used to attribute data. To achieve this, we traverse the training samples by calculating the IS value for each sample to select the most valuable data. We then remove these samples and update the model through retraining or using the SAM-HIF (Fast), SAM-HIF, and SAM-GIF, and analyzing the changes in accuracy relative to the accuracy before removal. In detail, we first calculate the influence scores of all samples and select the

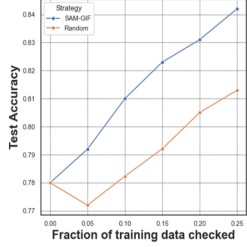 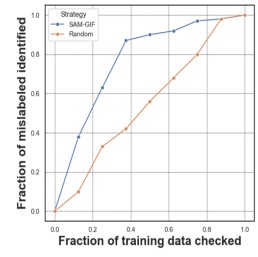

(a) Test Accuracy      (b) Identified Ratio

Figure 2: Harmful Data Removal Experiment.

most valuable ones by ranking them. Then, the top k most valuable data points are removed, where k ranges from 2% to 10% of the data size. Next, the model is retrained multiple times using different random seeds or updated using our different methods. The test set accuracy is then calculated after updating the parameters. Additionally, we randomly delete the same proportion of samples as a control experiment. Our experimental results, as shown in the Figure 1, indicate that after removing the most valuable training data, the accuracy of the updated model decreases across different datasets. For instance, for the CIFAR-10 dataset, when the removal ratios are 0.02, 0.05, 0.07, and 0.10, the accuracy scores for retraining and the SAM-HIF (Fast), SAM-HIF, and SAM-GIF algorithms gradually decrease. At the 0.10 removal ratio, the accuracy for the four methods drops to 0.8432, 0.8212, 0.8412, and 0.8311, respectively. The accuracy of different approaches is similar to that of retraining (with a maximum difference of 0.022), which demonstrates that our algorithm significantly reduces computational time while maintaining accuracy. In contrast, random deletion exhibits greater variability, and although a higher deletion ratio can improve accuracy, the results are less stable.

## 5.5 RESULTS OF IDENTIFYING HARMFUL DATA

To evaluate our algorithm's ability to identify harmful data, we introduce noise into CIFAR-10 by randomly inverting labels for 10% of the training set. Models are trained with SAM using the same setup as previous experiments. We compute each sample's IS with SAM-GIF and label samples with low IS as harmful. We then retrain models while removing increasing proportions of these

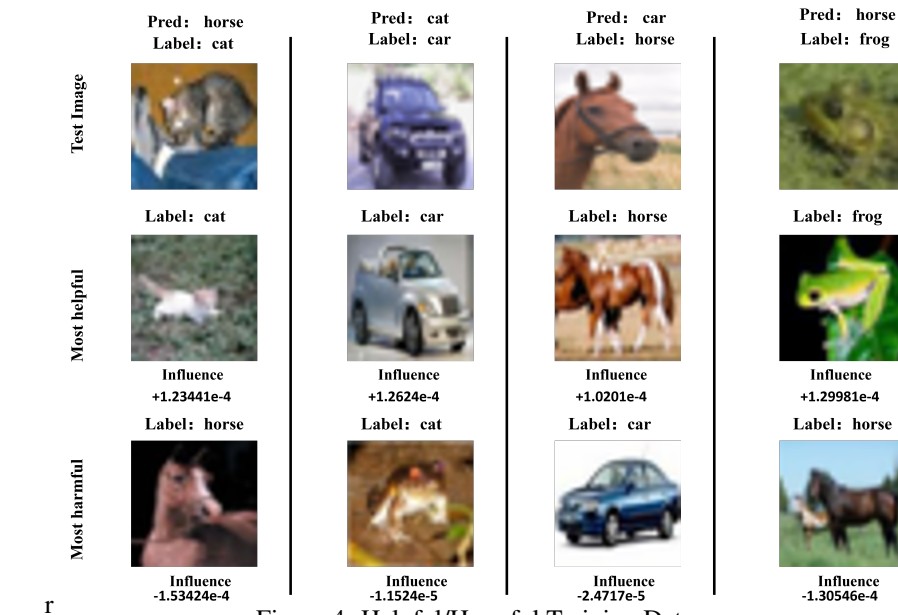

r

Figure 4: Helpful/Harmful Training Data

harmful samples and report test accuracy (Figure 2a). As more harmful data are removed, accuracy improves; at a 25% removal rate, accuracy increases by about 0.06. In contrast, random removal leads to unstable or reduced accuracy. As shown in Figure 2b, our method detects over 90% of noisy data when the detection rate reaches 0.4, outperforming random deletion. Results on additional datasets and baseline comparisons are provided in the Appendix D.2. We further conduct the same harmful data removal experiments on the CUB and FOOD-101 datasets, as shown in Figure 3.

As the removal ratio increases, the accuracy of all methods improves, but our methods (SAM-GIF and SAM-HIF) consistently outperform the baseline approaches, including TARK, IF, and IF-EKFAC. Notably, when 20% of the data is removed, SAM-GIF achieves about 65.5% accuracy on CUB and around 76.5% on FOOD-101, which are both higher than those achieved by the baseline methods.

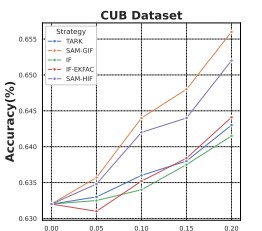 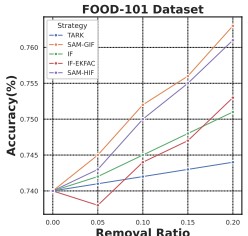

Figure 3: Baselines for Harmful Data Removal

### 5.6 RESULTS ON INTERPRETABILITY

We apply SAM-GIF to identify the training samples that are most relevant to specific prediction errors in the test samples. In this process, we select samples from the test data where the model misclassifies. By calculating the IS of the samples, we identify and visualize harmful data with a negative IS that leads to erroneous predictions, namely. Figure 4 presents the visualization of this error prediction tracing process for the CIFAR-10 dataset (more visualization results are in Appendix). The first row displays examples of misclassified test samples, the second row shows the most influential training data for classifying this sample, and the third row shows the most harmful training data for this classification. We can trace the outcomes of the error prediction process through the visualization results. Please see Appendix D.3 for more results. We also conduct an ablation study (see Appendix D.4).

## 6 CONCLUSION

In conclusion, we addressed data attribution challenges in the SAM framework with two novel methods: SAM-HIF and SAM-GIF. SAM-HIF employs a comprehensive closed-form data influence estimation. SAM-GIF utilizes gradient trajectory information for efficient, scalable evaluation. Experiments on four datasets indicated the effectiveness and scalability of these methods. Our approach aids in mislabeled data detection, model editing, and interpretability in SAM-trained models.

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

## A    THE USE OF LARGE LANGUAGE MODELS (LLMS)

We used LLMs to refine grammar and improve language fluency. The authors reviewed and edited all LLM-generated content and assume full responsibility for the final text.

## B    OMITTED PROOFS

### B.1    DATA ATTRIBUTION IN SAM VIA HESSIAN IF

**Theorem B.1.** *Consider the data $(x_k, y_k)$ along with a SAM-trained model $\omega^*$, $\hat{\epsilon}(\omega^*)$ represents the model weight perturbation that results in the largest loss. Then the corresponding SAM-IF is defined as:*

$$\textit{SAM-IF}(x_k, y_k) = -H_\omega^{-1} \cdot \nabla L_S^k(\omega^* + \hat{\epsilon}(\omega^*))$$

*where $H_\omega$ is defined as $H_\omega = \nabla^2 L_S(\omega^* + \hat{\epsilon}(\omega^*)) + \lambda \cdot I$. $\omega_k$ can be approximated by*

$$\omega_k \approx \omega^* - \textit{SAM-IF}(x_k, y_k).$$

*Proof.* Recalling the definition $\hat{\epsilon}(\omega) = \arg\max_{||\epsilon||_p \leq \rho} L_S(\omega + \epsilon)$ and

$$
\begin{aligned}
\omega^* &= \arg\min_\omega L_S(\omega) + \frac{\lambda}{2} \cdot ||\omega||_2^2 \\
&= \arg\min_\omega L_S(\omega + \hat{\epsilon}(\omega)) + \frac{\lambda}{2} \cdot ||\omega||_2^2 \\
&= \arg\min_\omega \sum_{i=1}^n \ell(x_i, y_i; \omega + \hat{\epsilon}(\omega)) + \frac{\lambda}{2} \cdot ||\omega||_2^2 \\
&= \arg\min_\omega \sum_{i=1}^n L_S^i(\omega + \hat{\epsilon}(\omega)) + \frac{\lambda}{2} \cdot ||\omega||_2^2
\end{aligned}
\tag{6}
$$

Now we consider removing the $k$-th data $(x_k, y_k)$. The retrained parameter after the removal is denoted as $\omega_k$:

$$\omega_k = \arg\min_\omega \sum_{i=1, i \neq k}^n L_S^i(\omega + \hat{\epsilon}(\omega)) + \frac{\lambda}{2} \cdot ||\omega||_2^2$$

To estimate this $\omega_k$, we can firstly up-weigh the loss term $L_S^k(\omega + \hat{\epsilon}(\omega))$ by $\delta$ in the original loss function defined in (6), then a series of optimization problem can be defined as:

$$\omega_\delta = \arg\min_\omega L_S(\omega + \hat{\epsilon}_\delta(\omega)) + \delta \cdot L_S^k(\omega + \hat{\epsilon}_\delta(\omega)) + \frac{\lambda}{2} \cdot ||\omega||_2^2 \tag{7}$$

Noting here $\hat{\epsilon}_\delta(\omega)$ will change according to $\delta$ and is defined as

$$\hat{\epsilon}_\delta(\omega) = \arg\max_{||\epsilon||_p \leq \rho} \left[ L_S(\omega + \epsilon) + \delta \cdot L_S^k(\omega + \epsilon) \right]$$

From the minimizing condition in Equation (6) and (7) and Lemma 4.2, we have

$$\nabla L_S(\omega^* + \hat{\epsilon}(\omega^*)) = 0, \nabla L_S(\omega_\delta + \hat{\epsilon}_\delta(\omega_\delta)) + \delta \cdot \nabla L_S^k(\omega_\delta + \hat{\epsilon}_\delta(\omega_\delta)) = 0. \tag{8}$$

To estimate $\omega_\delta - \omega^*$, we perform a Taylor expand at $\omega^*$ for the first equation in (8).:

$$
\begin{aligned}
0 = \nabla L_S\left(\omega^* + \hat{\epsilon}(\omega^*)\right) + \delta \cdot \nabla L_S^k\left(\omega^* + \hat{\epsilon}(\omega^*)\right) + \nabla^2 L_S\left(\omega^* + \hat{\epsilon}(\omega^*)\right) \cdot (\omega_\delta - \omega^*) \\
+ \nabla^2 L_S\left(\omega^* + \hat{\epsilon}(\omega^*)\right) \cdot \left(\hat{\epsilon}_\delta\left(\omega_\delta\right) - \hat{\epsilon}(\omega^*)\right)
\end{aligned}
\tag{9}
$$

In the above expansion, the first term equals 0 from (8). And to make thing easier, we can neglect the last term and deduce a simple influence function for SAM.

$$\text{SAM-IF}(x_k, y_k) = \left. \frac{\mathrm{d}\omega_\delta}{\mathrm{d}\delta} \right|_{\delta=0} = -H_\omega^{-1} \cdot \nabla L_S^k(\omega^* + \hat{\epsilon}(\omega^*)),$$

where $H_\omega$ is defined as $H_\omega = \nabla^2 L_S\left(\omega^* + \hat{\epsilon}(\omega^*)\right) + \lambda \cdot I$.

When $\delta = -1$, $\omega_\delta$ becomes $\omega_k$. Then $\omega_k$ can be approximated by a first-order Taylor expansion:

$$\omega_k \approx \omega^* - \text{SAM-IF}(x_k, y_k).$$

$\square$

**Lemma B.2.** *The term $\hat{\epsilon}_{k,\delta}\left(\omega_{k,\delta}\right) - \hat{\epsilon}(\omega^*)$ can be approximated by*

$$
\begin{aligned}
&\hat{\epsilon}_{k,\delta}(\omega_{k,\delta}) - \hat{\epsilon}(\omega^*) \\
=&\hat{\epsilon}_{k,\delta}(\omega^*) - \hat{\epsilon}(\omega^*) + \left.\frac{\mathrm{d}\hat{\epsilon}_{k,\delta}(\omega)}{\mathrm{d}\omega}\right|_{\omega=\omega^*} \cdot (\omega_{k,\delta} - \omega^*).
\end{aligned}
\tag{10}
$$

*Proof.* Now, we begin to estimate $\hat{\epsilon}_{k,\delta}(\omega_{k,\delta}) - \hat{\epsilon}(\omega^*)$. Recalling the definitions as following:

$$
\hat{\epsilon}_{k,\delta}\left(\omega_{k,\delta}\right) = \underset{||\epsilon||_p \leq \rho}{\arg\max}\left[L_S(\omega_{k,\delta} + \epsilon) + \delta \cdot L_S^k(\omega_{k,\delta} + \epsilon)\right],
$$

$$
\hat{\epsilon}(\omega^*) = \underset{||\epsilon||_p \leq \rho}{\arg\max}.L_S(\omega^* + \epsilon)
$$

We will use $\hat{\epsilon}_{k,\delta}(\omega^*)$ as an intermediate variable to simplify the approximation, which is defined as

$$
\hat{\epsilon}_{k,\delta}(\omega^*) = \underset{||\epsilon||_p \leq \rho}{\arg\min}\left[L_S(\omega^* + \epsilon) + \delta \cdot L_S^k(\omega^* + \epsilon)\right]
$$

Firstly, we perform a Taylor expansion for $\hat{\epsilon}_{k,\delta}\left(\omega_{k,\delta}\right)$ at $\omega^*$ as follows:

$$
\hat{\epsilon}_{k,\delta}\left(\omega_{k,\delta}\right) \approx \hat{\epsilon}_{k,\delta}\left(\omega^*\right) + \left.\frac{\mathrm{d}\hat{\epsilon}_{k,\delta}\left(\omega_{k,\delta}\right)}{\mathrm{d}\omega}\right|_{\omega=\omega^*} \cdot (\omega_{k,\delta} - \omega^*)
$$

Then the objective to estimate becomes

$$
\begin{aligned}
&\hat{\epsilon}_{k,\delta}(\omega_{k,\delta}) - \hat{\epsilon}(\omega^*) \\
=&\hat{\epsilon}_{k,\delta}(\omega_{k,\delta}) - \hat{\epsilon}_{k,\delta}(\omega^*) + \hat{\epsilon}_{k,\delta}(\omega^*) - \hat{\epsilon}(\omega^*) \\
=&\hat{\epsilon}_{k,\delta}(\omega^*) - \hat{\epsilon}(\omega^*) + \left.\frac{\mathrm{d}\hat{\epsilon}_{k,\delta}\left(\omega_{k,\delta}\right)}{\mathrm{d}\omega}\right|_{\omega=\omega^*} \cdot (\omega_{k,\delta} - \omega^*)
\end{aligned}
$$

$\square$

**Theorem B.3.** *Consider the $k$-th data $(x_k, y_k)$ along with a SAM-trained model $\omega^*$. Define the Hessian-based influence function for SAM(SAM-HIF) as:*

$$
\begin{aligned}
&SAM\text{-}HIF(x_k, y_k) \\
=&-\left(H_\omega + H_\omega \cdot \frac{\mathrm{d}\hat{\epsilon}(\omega^*)}{\mathrm{d}\omega}\right)^{-1} \cdot \nabla L_S^k\left(\omega^* + \hat{\epsilon}(\omega^*)\right).
\end{aligned}
$$

*Then $\omega_k$ can be approximated by:*

$$\omega_k \approx \omega^* - SAM\text{-}HIF(x_k, y_k).$$

*Proof.* Based on the expansion in Equation (9) in Theorem B.1, we can further estimate the term $\hat{\epsilon}_{k,\delta}\left(\omega_{k,\delta}\right) - \hat{\epsilon}(\omega^*)$ by Lemma B.2:

$$
\begin{aligned}
0 =& \nabla L_S\left(\omega^* + \hat{\epsilon}(\omega^*)\right) + \delta \cdot \nabla L_S^k\left(\omega^* + \hat{\epsilon}(\omega^*)\right) + \nabla^2 L_S\left(\omega^* + \hat{\epsilon}(\omega^*)\right) \cdot (\omega_\delta - \omega^*) \\
&+ \nabla^2 L_S\left(\omega^* + \hat{\epsilon}(\omega^*)\right) \cdot \left(\hat{\epsilon}_\delta(\omega^*) - \hat{\epsilon}(\omega^*) + \left.\frac{\mathrm{d}\hat{\epsilon}_\delta\left(\omega_\delta\right)}{\mathrm{d}\omega}\right|_{\omega=\omega^*} \cdot (\omega_\delta - \omega^*)\right)
\end{aligned}
\tag{11}
$$

The first term equals 0 from (8). Then we have

$$
0 = \delta \cdot \nabla L_S^k\left(\omega^* + \hat{\epsilon}(\omega^*)\right) + \left(H_\omega + H_\omega \cdot \frac{\mathrm{d}\hat{\epsilon}_\delta\left(\omega_\delta\right)}{\mathrm{d}\omega}\right) \cdot (\omega_\delta - \omega^*) + H_\omega \cdot \left(\hat{\epsilon}_\delta(\omega^*) - \hat{\epsilon}(\omega^*)\right).
\tag{12}
$$

Then

$$\omega_\delta - \omega^* = -\left(H_\omega + H_\omega \cdot \frac{\mathrm{d}\hat{\epsilon}_\delta\left(\omega_\delta\right)}{\mathrm{d}\omega}\right)^{-1} \cdot H_\omega \cdot \left(\hat{\epsilon}_\delta(\omega^*) - \hat{\epsilon}(\omega^*)\right)$$

$$- \delta \cdot \left(H_\omega + H_\omega \cdot \frac{\mathrm{d}\hat{\epsilon}_\delta\left(\omega_\delta\right)}{\mathrm{d}\omega}\right)^{-1} \cdot \nabla L_S^k\left(\omega^* + \hat{\epsilon}(\omega^*)\right).$$

Then we can obtain the following equation:

$$\frac{\mathrm{d}\omega_\delta}{\mathrm{d}\delta}\bigg|_{\delta=0} = -\left(H_\omega + H_\omega \cdot \frac{\mathrm{d}\hat{\epsilon}\left(\omega^*\right)}{\mathrm{d}\omega}\right)^{-1} \cdot H_\omega \cdot \frac{\mathrm{d}\hat{\epsilon}_\delta(\omega^*)}{\mathrm{d}\delta}\bigg|_{\delta=0}$$

$$- \left(H_\omega + H_\omega \cdot \frac{\mathrm{d}\hat{\epsilon}\left(\omega^*\right)}{\mathrm{d}\omega}\right)^{-1} \cdot \nabla L_S^k\left(\omega^* + \hat{\epsilon}(\omega^*)\right).$$

To enhance the computation efficiency, we drop the first term, and obtain the SAM-HIF as

$$\text{SAM-IF} = \frac{\mathrm{d}\omega_\delta}{\mathrm{d}\delta}\bigg|_{\delta=0} = -\left(H_\omega + H_\omega \cdot \frac{\mathrm{d}\hat{\epsilon}\left(\omega^*\right)}{\mathrm{d}\omega}\right)^{-1} \cdot \nabla L_S^k\left(\omega^* + \hat{\epsilon}(\omega^*)\right). \tag{13}$$

$\square$

## B.2 Computation Acceleration

**Theorem B.4.** *Given a validation dataset defined as $D_{val} = \{(x_t, y_t)\}_{t=1}^n$. Denote the SAM-retrained model after the removal of $(x_k, y_k)$ as $\omega_{-k}^*$, then*

$$\sum_{(x,y)\in D_{val}} \ell(x, y; \omega^*) - \sum_{(x,y)\in D_{val}} \ell(x, y; \omega_{-k}^*)$$

$$\approx \sum_{(x,y)\in D_{val}} \nabla\ell(x, y; \omega^*) \cdot IF(x, y) \triangleq IS(x, y),$$

*where IF can be SAM-HIF or SAM-GIF. We define the right hand of the above equation as the Influence Score (IS).*

*Proof.*

$$\sum_{(x,y)\in D_{val}} \ell(x, y; \omega^*) - \sum_{(x,y)\in D_{val}} \ell(x, y; \omega_{-k}^*)$$

$$\approx \sum_{(x,y)\in D_{val}} \nabla\ell(x, y; \omega^*) \cdot \left(\omega^* - \omega_{-k}^*\right)$$

$$= \sum_{(x,y)\in D_{val}} \nabla\ell(x, y; \omega^*) \cdot \text{IF}(x, y) \triangleq \text{IS}(x, y).$$

$\square$

## C Omitted Algorithms

---

**Algorithm 1** SAM Algorithm in (Foret et al., 2020)

---

**Require:** Training set $S$, Loss function $L$, Batch size $b$, Step size $\eta > 0$, Neighborhood size $\rho > 0$.
**Ensure:** Model trained with SAM
1: Initialize weights $w_0$, $t = 0$;
2: **while** not converged **do**
3:     Sample batch $B = \{(x_1, y_1), \ldots, (x_b, y_b)\}$;
4:     Compute gradient $\nabla_w L_{\mathcal{B}}(w_t)$ of batch's training loss;
5:     Compute $\hat{\epsilon}(w)$ per Equation (4);
6:     Compute gradient: $g = \nabla_w L_B(w_t + \hat{\epsilon}(w_t))$;
7:     Update weights: $w_{t+1} = w_t - \eta g$; $t = t + 1$;
8: **end while**

---

**Algorithm 2** SAM-IF with gradient trajectory

---

1: **Input:**
    **data:** Training Dataset $S = \{(x_i, y_i)\}_{i=1}^n$, data point $(x_k, y_k)$ to be evaluated.
    **Parameter:** Parameter checkpoint set $\Omega = \{\omega_c\}_{c=1}^s$, learning rate $\eta_c$ at step $c$.
2: Compute the influence of $(x_k, y_k)$ in the $c$-th checkpoint as

$$\text{IF}_c = \eta_c \cdot \nabla L_S^k(\omega_{c-1} + \hat{\epsilon}(\omega_{c-1}))$$

3: Sum up to obtain the final influence as

$$\text{SAM-IF}_{\text{Step}} = \sum_{c=1}^s \text{IF}_c$$

4: **Return:** SAM-IF$_{\text{Step}}$. Final Parameter $\omega^- = \omega^* - \text{SAM-IF}$.

---

**Algorithm 3** Simple SAM-IF

---

1: **Input:**
    **Data:** Training Dataset $S = \{(x_i, y_i)\}_{i=1}^n$, data point $(x_k, y_k)$ to be evaluated.
    **Parameter:** The learned SAM-parameter $\omega^*$, the learned best perturbation $\hat{\epsilon}$.
2: Compute $T^k$ as:
$$T^k = \nabla L_S^k(\omega^* + \hat{\epsilon})$$

3: Define the Hessian matrix $H$ as:
$$\nabla^2 L_S(\omega^* + \hat{\epsilon})$$

4: Use EK-FAC to compute the Hessian-vector product of $H$ and $T^k$:
$$\text{SAM-IF} = -H^{-1} \cdot T^k$$

5: Obtain the estimated parameter by
$$\omega^- = \omega^* - \text{SAM-IF}.$$

6: **Return:** SAM-IF, $\omega^-$.

---

---

**Algorithm 4** SAM-IF with total Hessian

---

1: **Input:**
   **Data:** Training Dataset $S = \{(x_i, y_i)\}_{i=1}^n$, data point $(x_k, y_k)$ to be evaluated.
   **Parameter:** The learned SAM-parameter $\omega^*$, the learned best perturbation $\hat{\epsilon}$.
2: Compute $T^k$ as:
$$T^k = \nabla L_S^k (\omega^* + \hat{\epsilon})$$

3: Define the gradient of $\epsilon$ as
$$C^\epsilon = \nabla_\omega \frac{|\nabla_w L_S(\omega^*)|^{q-1}}{(\|\nabla_w L_S(\omega^*)\|_q^q)^{1/p}}$$

4: Give the definition of $H$ as
$$H = \nabla^2 L_S (\omega^* + \hat{\epsilon}) + \nabla^2 L_S (\omega^* + \hat{\epsilon}) \cdot C^\epsilon.$$

5: $j \leftarrow 1$
   $T^k \leftarrow I_0$
6: **if** $\|I_j - I_{j-1}\|_1 > \zeta$ **then**
7:    Use EK-FAC to compute the Hessian-vector product $S_j \triangleq H \cdot I_j$.
8:    Compute $I_{j+1}$ by
$$I_{j+1} = I_j - \delta \cdot I_j + S_j + T^k$$

9:    $j \leftarrow j + 1$
10: **end if**
11: SAM-IF $\leftarrow I_{j+1}$.
12: Obtain the estimated parameter by $\omega^- = \omega^* - $ SAM-IF.
13: **Return:** SAM-IF. $\omega^-$.

---

# D   ADDITIONAL EXPERIMENT RESULTS

This section presents the additional experimental results for completeness and detailed analysis.

## D.1   RESULTS OF EFFICIENCY AND ACCURACY

To further validate the effectiveness and generalizability of our approach, we conducted extensive experiments across a range of datasets (MNIST, HAM, CUB, FOOD-101) and models (Wide-Resnet (Zagoruyko & Komodakis, 2016), ViT (Dosovitskiy et al., 2020), and ResNet50 (He et al., 2016)).

We first evaluate on MNIST and HAM datasets using Wide-Resnet. Beyond these, we extend our comparison to the CUB and FOOD-101 datasets with both ViT and ResNet50 architectures. For each setting, we compare the proposed SAM-GIF with baseline methods, including Retrain, SAM-HIF, and SAM-HIF(Fast), reporting both test accuracy and runtime (RT in seconds).

Table 2: Performance comparison on MNIST and HAM.

| Method | Mnist | | HAM | |
|---|---|---|---|---|
| | Accuracy | RT (second) | Accuracy | RT (second) |
| Retrain | 0.9927±0.0005 | 2304.24±7.91 | 0.7254±0.02 | 2300.00±10.00 |
| SAM-HIF(Fast) | 0.9876±0.0013 | 8.0698±2.91 | 0.7130±0.0150 | 15.00±2.00 |
| SAM-HIF | 0.9880±0.0011 | 18.2321±1.91 | 0.7212±0.0120 | 52.20±3.00 |
| SAM-GIF | 0.9884±0.0007 | 2.8212±1.91 | 0.7350±0.0110 | 4.320±1.50 |

Table 3: Performance comparison on *ViT*.

| Method | CUB | | FOOD-101 | |
|---|---|---|---|---|
| | Accuracy | RT (s) | Accuracy | RT (s) |
| Retrain | 0.7236±0.005 | 2876.23±12.47 | 0.8231±0.031 | 16310.24±25.32 |
| SAM-HIF | 0.7032±0.005 | 53.2310±3.78 | 0.7932±0.024 | 47.212±4.12 |
| SAM-HIF(Fast) | 0.7189±0.003 | 22.3211±5.16 | 0.8012±0.014 | 28.31±3.21 |
| SAM-GIF | 0.7214±0.003 | 14.3231±2.33 | 0.8123±0.047 | 15.214±3.23 |

Table 4: Performance comparison on *Resnet50*.

| Method | CUB | | FOOD-101 | |
|---|---|---|---|---|
| | Accuracy | RT (s) | Accuracy | RT (s) |
| Retrain | 0.6336±0.005 | 2197.33±12.47 | 0.7436±0.031 | 15900.24±25.32 |
| SAM-HIF | 0.6213±0.005 | 37.2590±3.78 | 0.7022±0.024 | 41.321±4.12 |
| SAM-HIF(Fast) | 0.6178±0.003 | 25.3214±5.16 | 0.6918±0.014 | 26.0698±3.21 |
| SAM-GIF | 0.6244±0.004 | 12.6789±2.33 | 0.7189±0.047 | 13.234±3.23 |

Across all datasets and model architectures, SAM-GIF achieves test accuracy very close to, or even surpassing, that of full retraining. For example, on the HAM dataset (Wide-Resnet), SAM-GIF reaches 0.7350 accuracy, outperforming Retrain (0.7254). On CUB and FOOD-101, both with ViT and ResNet50, SAM-GIF maintains competitive accuracy relative to all baselines.

SAM-GIF consistently offers significant speedup over retraining. For instance, on MNIST and HAM with Wide-Resnet, the runtime of SAM-GIF is just 2.8s and 4.3s respectively, compared to over 2300s for retraining—about three orders of magnitude faster. Similar trends hold for ViT and ResNet50, where SAM-GIF achieves the lowest runtime among all methods.

Overall, these results demonstrate that SAM-GIF achieves an excellent balance, providing high accuracy nearly on par or superior to retraining, while drastically reducing computational costs across diverse datasets and model architectures.

## D.2   RESULTS OF IDENTIFYING HARMFUL DATA

We conduct additional harmful data identification experiments on the MNIST, CIFAR-100, and Mini-ImageNet datasets. The results are listed as follows. Figures 5, 6, and 7 illustrate the effec-

tiveness of the SAM-GIF algorithm in detecting and removing harmful data. The superiority of our method is demonstrated across multiple datasets. We observe that, compared to random removal, the harmful detection rate of the SAM-GIF algorithm reaches approximately 80% at a removal rate of 0.4. Moreover, by removing harmful data, the model's accuracy gradually improves.

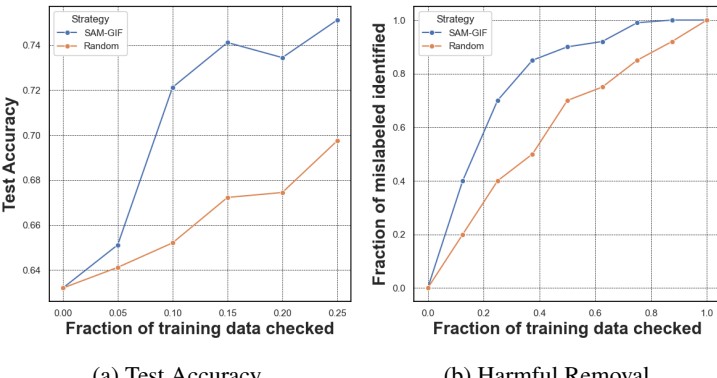

(a) Test Accuracy          (b) Harmful Removal

Figure 5: Harmful data removal experiment on CIFAR-100 dataset. IS: using the influence score to determine which sample to remove. Random: randomly removing tasks.

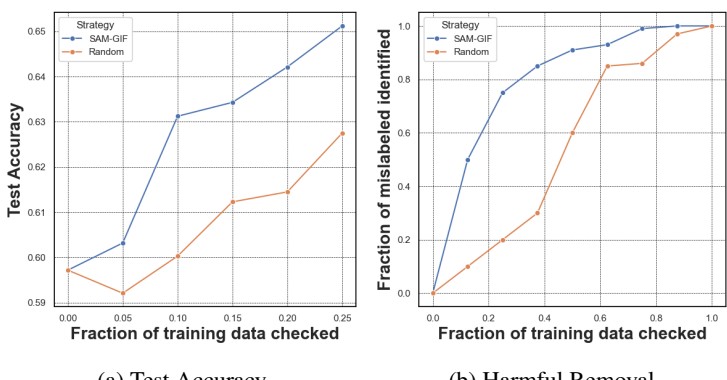

(a) Test Accuracy          (b) Harmful Removal

Figure 6: Harmful Removal on Mini-ImageNet. IS: using the influence score to determine which sample to remove. Random: randomly removing tasks.

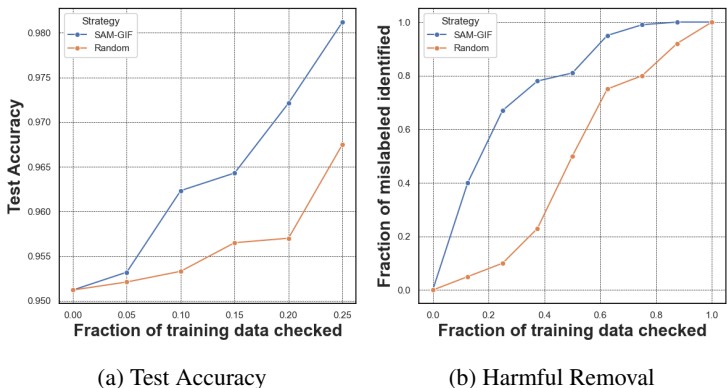

(a) Test Accuracy          (b) Harmful Removal

Figure 7: Harmful data removal experiment on MNIST dataset. IS: using the influence score to determine which sample to remove. Random: randomly removing tasks

### D.3 ADDITIONAL RESULTS ON INTERPRETABILITY

Figures 8, 9, and 10 present additional visualization results of the error prediction tracing process on the MNIST, CIFAR-100, and Mini-ImageNet datasets, respectively. The first row displays examples

of misclassified test samples, the second row shows the most influential training data for classifying these samples, and the third row presents the most harmful training data for these classifications. These visualization results allow us to trace the outcomes of the error prediction process effectively.

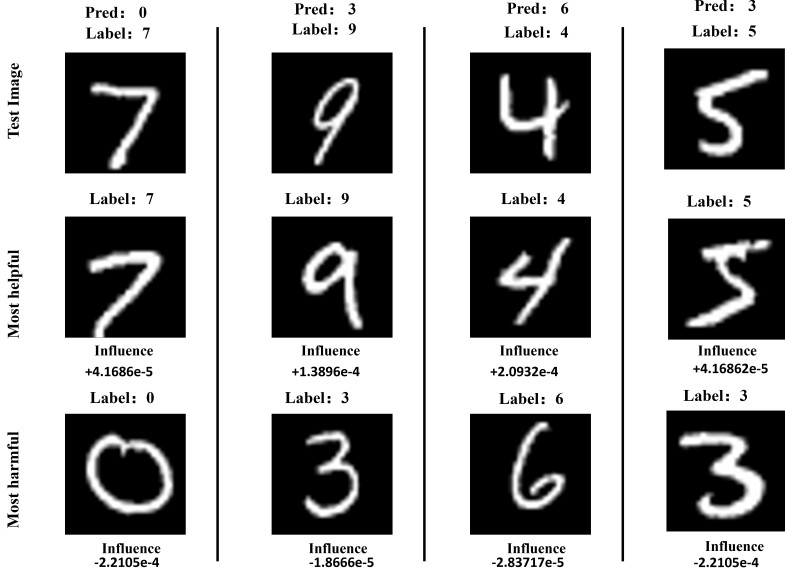

Figure 8: The most helpful and harmful training data tracked by misclassified data on MNIST dataset

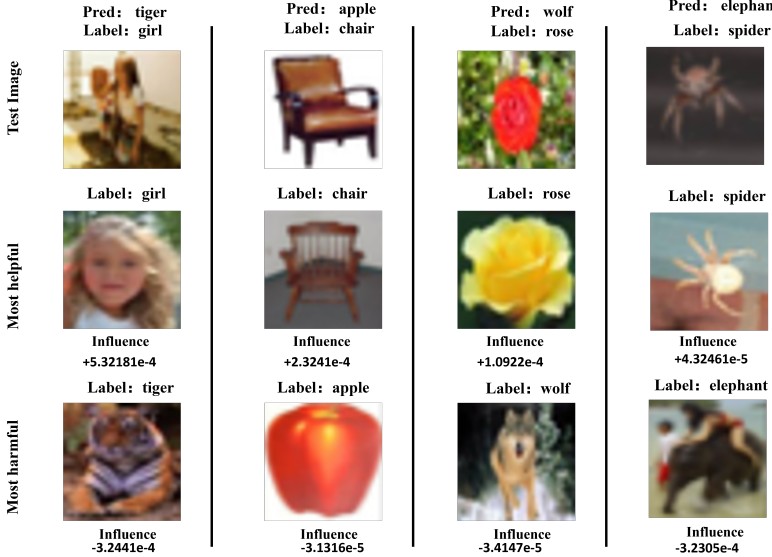

Figure 9: The most helpful and harmful training data tracked by misclassified data on CIFAR-100 dataset

## D.4 ADDITIONAL ABLATION STUDY

## D.5 ABLATION STUDY

We conducted ablation experiments on above three methods, SAM-HIF (fast), SAM-HIF and SAM-GIF. We randomly removed 1%-8% of the training samples from CIFAR-10 and CIFAR-100 datasets, and evaluated the model parameters using SAM-HIF(fast) and SAM-HIF, with retrain serving as the ground truth. The results, shown in Figure 11, clearly demonstrate that SAM-HIF consistently outperforms SAM-HIF(Fast) during the process of data removal. Also, we can see

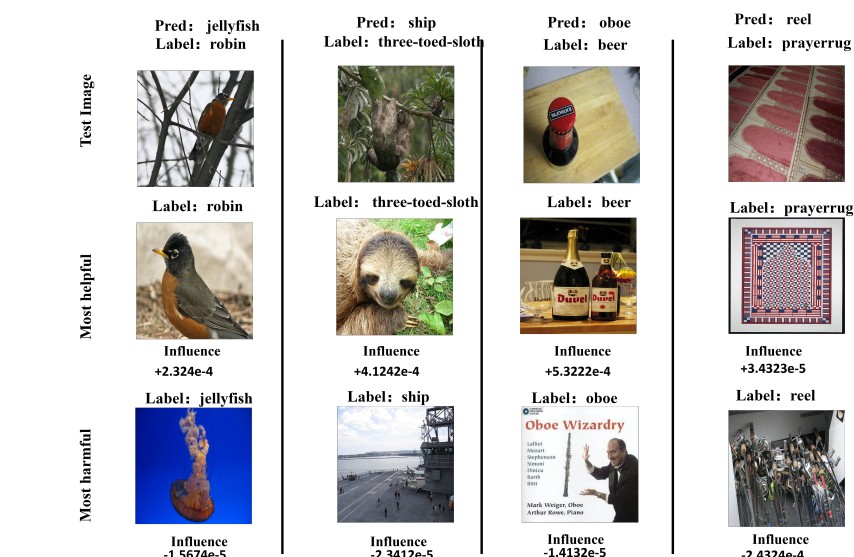

Figure 10: The most helpful and harmful training data tracked by misclassified data on Mini-ImageNet dataset

both methods consistently have slight differences with the ground truth. Results for SAM-GIF are included in the Appendix D.4.

Figure 12 shows the ablation study results of SAM-GIF. We primarily tested the impact of different numbers of checkpoint weights on the SAM-GIF algorithm using the CIFAR-10 dataset. From the figure, we can observe that as the number of checkpoints increases, the accuracy of SAM-GIF becomes closer to that of retraining. When the number of checkpoints is 10, the accuracy of SAM-GIF is 0.9497, while the retraining accuracy is 0.9517. At the same time, as the number of checkpoints increases, the running time of SAM-GIF also increases.

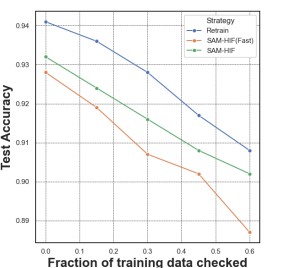 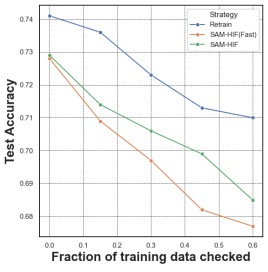

(a) CIFAR-10        (b) CIFAR-100

Figure 11: Ablation studies on CIFAR-10 and CIFAR-100.

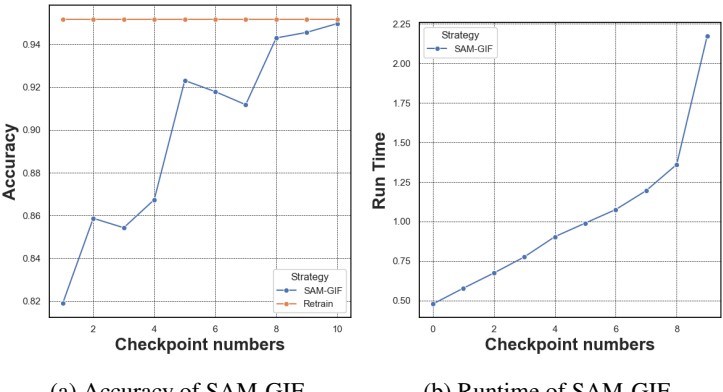

(a) Accuracy of SAM-GIF      (b) Runtime of SAM-GIF

Figure 12: Ablation study of SAM-GIF on Cifar10 dataset

