# OpenReview forum: "Attributing Data for Sharpness-Aware Minimization"
_ICLR.cc/2026/Conference — ICLR 2026 Conference Withdrawn Submission_

### Official Review · Reviewer_TEHh · 2025-10-17

**Soundness:** 2
**Presentation:** 2
**Contribution:** 1
**Rating:** 4
**Confidence:** 4

**Summary:**

This work aims to extend the application of Influence Functions (IF) to the Sharpness-Aware Minimization (SAM) framework. The authors first derive SAM-IF, an adaptation of IF for the SAM setting. They then propose two variants, SAM-HIF and SAM-GIF, which compute influence values based on the Hessian and the gradient trajectory, respectively. Finally, they evaluate the proposed methods across different tasks.

**Strengths:**

1. Exploring the effectiveness of Influence Functions (IF) within the SAM framework is a relatively novel direction.

**Weaknesses:**

1. The contribution of this paper appears limited. The main contribution lies in adapting the Influence Function (IF) framework to the SAM setting, while the derivation closely follows the methodology used in the original IF paper. Moreover, the idea of leveraging gradient trajectories has already been explored in TracIn [1].
2. It is unclear why the authors compare their proposed method with Retrain in all experiments related to efficiency and accuracy. Since the main motivation of this paper is that the original Influence Function (IF) cannot be directly applied to the SAM setting, the comparison should instead focus on SAM-IF versus the original IF and TracIn [1]. Such comparisons would more appropriately demonstrate the effectiveness of the proposed method within the SAM framework.
3. Similar to Weakness 2, the experiments on harmful data removal lack key baselines, specifically, the original IF and TracIn, across the CIFAR-10, CIFAR-100, Mini-ImageNet, and MNIST datasets.
4. The generalization ability of the proposed method across different SAM variants lacks sufficient discussion and experimental validation. There exist several variants of SAM, such as FisherSAM [1] and FriendlySAM [2], each exhibiting distinct optimization behaviors. However, the paper provides neither a theoretical analysis nor empirical evidence regarding whether the proposed approach can be effectively applied to these SAM variants.

Reference

[1] Estimating Training Data Influence by Tracing Gradient Descent, NeurIPS, 2020

[2] Fisher SAM: Information Geometry and Sharpness Aware Minimisation, ICML, 2022

[3] Friendly Sharpness-Aware Minimization, CVPR, 2024

**Questions:**

See Weaknesses.

---

### Official Review · Reviewer_19nk · 2025-10-30

**Soundness:** 3
**Presentation:** 4
**Contribution:** 2
**Rating:** 4
**Confidence:** 3

**Summary:**

This paper propose an extension to original influence function (IF) and make it suitable (more accurate) for Sharpness-Aware Minimization (SAM, a novel optimization method for ML models that requires bi-level structure optimization). This paper propose two innovative data valuation/data attribution methods for SAM the takes the change of perturbation term into consideration. Some empirical results shows the good effectiveness of SAM-specific data attribution.

**Strengths:**

- The motivation of this paper is clear. SAM as an important optimization method and the influence function to it is important to expand the applicability of data attribution
- The presentation (preliminary of SAM, IF; derivation of SAM-IF, SAM-HIF, and SAM-GIF) is clear enough and fluent for the reader.
- The results shows that the proposed SAM specific influence function is effective on several settings.

**Weaknesses:**

- The experiment design is somehow problematic.
  - The evaluation metric of the experiment seems lack clearance. For example, the "accuracy" shown in Figure 1 and Table 1 is somehow hard to interpret the effectiveness of SAM specific IFs. And the metric in Figure 2 (test accuracy after the removal of harmful training data (with flipped label)) could show the effectiveness while indirectly.
  - A suggestion to show this is to use a small model (could be linear) to show that the groundtruth (retrain LOO) is highly correlated (pearson correlation) with the result (scores) calculated by SAM specific IFs.
  - Lack of a simple baseline (SAM-IF)'s performance in the experiment. This could show the necessity to have this SAM specific IF derivation.

I am happy to raise my score if the experiment design is well explained and the pearson correlation  / spearman ranking correlation is shown for between the score and groundtruth.

**Questions:**

- Which scenario and use cases are SAM widely used? If there is a commonly accepted answer for this, an experiment on that use case could be useful to present the significance of the result.

---

### Official Review · Reviewer_GFBJ · 2025-10-31

**Soundness:** 3
**Presentation:** 2
**Contribution:** 2
**Rating:** 4
**Confidence:** 3

**Summary:**

Despite the widespread use of Sharpness-aware minimization (SAM) for improving generalization and the Influence function (IF) for measuring the value of each data instance across various fields, no research has been conducted considering them together. This study explores the IF for SAM, which takes the form of bi-level optimization, proposing SAM-HIF using the Hessian and SAM-GIF using gradient trajectories. Experimental results show the effectiveness and efficiency of the proposed framework.

**Strengths:**

1. Given the widespread use of SAM and IF, the efficient calculation of the influence function for SAM, the research question of this study, is both interesting and highly influential.

2. SAM-HIF and SAM-GIF are grounded in diverse theoretical foundations, and its derivation process is also straightforward to understand.

**Weaknesses:**

1. The mathematical expressions throughout the paper, including the proofs, lack completeness. $w_{k,\delta}$ and $w_{\delta}$, $e_{k,\delta}(w)$ and $e_{\delta}(w)$ are used interchangeably in the paper and proofs. The definition of $L_{S}^{SAM}$ includes a regularization term. Therefore, Eq.(3), which expresses the gradient of $L_{S}^{SAM}$, should include $\lambda w$. In Theorem 4.3, the Identity matrix $I$ is omitted from the definition of $H_w$. I believe the paper requires detailed review of the mathematical expressions.

2. Although the most logic relies on various approximations, the analysis of approximation error remains insufficient. First-order Taylor approximations are frequently employed in SAM-IF and SAM-HIF, while SAM-GIF omits the Hessian term to reduce computational complexity. Such widespread use of various approximations can significantly widen the gap between theory and experiment. Despite these risks, this study did not undertake a comprehensive and rigorous analysis of approximation errors.

3. In my opinion, SAM-IF and SAM-HIF are lack of novelty. Defining the maximizing perturbation vector and LOO retrained parameter in the SAM environment as Definition 4.1 is natural, but it represents a naïve extension of SAM and IF. Furthermore, the derivation process flow for SAM-IF and SAM-HIF differs little from that of the existing SAM and IF. Consequently, SAM-IF and SAM-HIF can be viewed as combining the two methodologies and expanding the equations; it is difficult to regard them as new theories or methodologies.

**Questions:**

1. I suggest to the authors whether lines 206–215 should be placed in the preliminary section, as they are not content pertaining to Definition 4.1 but rather results derivable from the original SAM.

2. I would like to know the theoretical and experimental basis for disregarding the final term in Eq.(9).

3. I am also curious whether SAM-IF, SAM-HIF and SAM-GIF are extendable to various SAM variants[1,2,3].

[1] ASAM: Adaptive Sharpness-Aware Minimization for Scale-Invariant Learning of Deep Neural Networks

[2] Efficient Sharpness-aware Minimization for Improved Training of Neural Networks

[3] Fisher SAM: Information Geometry and Sharpness Aware Minimisation

---

### Official Review · Reviewer_e8Wn · 2025-10-31

**Soundness:** 3
**Presentation:** 1
**Contribution:** 3
**Rating:** 4
**Confidence:** 3

**Summary:**

The authors consider the problem of data attribution for a different type of model optimization called sharpness aware minimazation (SAM), which is a method of training models that not only minimize loss, but are also expected to generalize well. The key challenge in data attribution for SAM is that SAM is a min-max optimization, whereas most existing data attribution methods focus on just minimization.

The leading approaches to data attribution are based on either "Hessian-based" or "gradient-based" influence functions. The classical Hessian-based attributions (such as IJ or TRAK) track the change in the local optima by taking a Hessian inverse times gradient, and the gradient based methods (such as MAGIC) track the effect of a sample on each step of the optimization trajectory by unrolling the training loop.

The authors derive extensions of both of these leading methods to SAM style problems, and evaluate them empirically over a wide range of usecases.

**Strengths:**

I do not have enough background in optimization to know how common SAM optimization is used vs the standard SGD / Adam.
Assuming that SAM is prevalent in practice, I think the results are interesting: they explore a new and unique challenge and the proposed methods seem to work well in practice.

**Weaknesses:**

Despite this, I think that the paper is lacking in its presentation -- to the point where I could not verify the main results and would not recommend its publication in ICLR in its current form.

The biggest problem by far is that the statements and assumptions of the theory section seem vague to me (especially if they are stated as "Theorems").
Moreover, the rest of the paper seems very rough, with both structural issues (lack of intuition / motivation for the theory, the main empirical results are relegated to a very small plot at the very end of the paper, etc.) and smaller issues (typos, inconsistent notation, etc.).

If the presentation of the theoretical and empirical results can be improved, that would greatly strengthen the paper.

Below is a more detailed list of my notes.


**Theory:**

The main theorems (4.3, 4.5, 4.6) give somewhat vague statements of the form $X$ can be estimated as $X \approx Y$, but, as far as I can tell, there are no clear conditions for the validity of this theorem / definition of what it means for $Y$ to be a good approximation.

Skimming through appendix B, it seems like these theorems are meant to state that these are the best or most natural definitions of SAM-IF / SAM-HIF / SAM-GIF (which would be interesting statements, but in my opinion, too vague to labeled as "theorems" or even "conjectures").

Or maybe the claim is that they are valid linear extrapolations of the effect of reweighting a sample?
If this is indeed what is meant by the theorems, then it should be stated more explicitly.

Moreover, it was hard for me to get intuition for the meanings of the theorems / their proofs from the main text of the paper (see concrete question in the questions section).

Finally, the theory section contains many expressions of the form $\nabla L(\omega + \epsilon(\omega))$, and it is unclear to me if the derivative $\nabla$ is taken over $\omega$, and if so, does it take into account the dependence of $\epsilon$ on $\omega$ in this derivation? In other words, if we defined the function $f: \omega \rightarrow L(\omega + \epsilon(\omega))$, would the expression mean the gradient of $f$ at $\omega$, or is the expression intended to mean that we take the derivative of the function $g:\omega \rightarrow L(\omega)$ and evaluate the derivative of $g$ at the point $\omega + \epsilon(\omega)$?

**Experiments:**

I found the experimental section of the paper very interesting, and think that it could be a very good motivation for the rest of the paper.
If I understood it correctly, the first experiments set the stage by showing that HIF/GIF approximations are much faster to compute than full retrains, while sacrificing only a little bit of accuracy, and this is followed up by Figure 3, which compares HIF and GIF to existing data attribution methods.

It seems to me that the first experiments are somewhat unsurprising (we expect IF to be faster than retraining), so while it makes sense to include them somewhere as a sanity check, the main focus should be on Figure 3, since this figure is in some sense the main result of this paper, as it shows that the proposed methods outperform existing approaches. But this result is somewhat "hidden" in a very small and hard to read figure at the end of the paper. Moving this result much earlier in the paper, making the figure larger and improving the styling (see some notes below) would draw more focus to this central result.

**Writing:**

In general, I feel like the current writing is very dense and hard to read, and was lacking a lot of intuition / clarity.

For instance, a clearer definition of SAM early on in the paper would enhance readability.

**Minor Comments / Corrections**

Below is a list of some more minor comments / corrections that I noted while reading the paper.

Lines 55-58: isn't this a somewhat restrictive definition of data attribution? In particular, this definition does not allow some applications like unlearning? How would this definition compare to the definition of as defined eg in the ICML 2024 data attribution workshop to be the problem of estimating the overall change in parameters / predictions?

Line 72-73: "...on model parameters, **is indirect manner** and has not been considered..."  -- typo / minor grammar error?

Line 212: "Follow the idea of influence function, we up-weigh the..." -- also a typo / minor grammar error?

Lines 240-250: It's not clear where this approximation comes from. Maybe consider adding some intuition?

Lines 290-300: this seems very similar to the approach of Koh and Liang 2017 "Understanding Black-box Predictions via Influence Functions", which uses a series of Hessian-vector inner products to compute the IF more efficiently. Could you please give some intuition on how your approach differs?

Lines 460-470: TARK should be TRAK, right?

Equation 5 (and I think also somewhere else): inconsistent notation ($\omega$ changes to $w$ and back) -- probably a typo.

**Questions:**

Beyond the questions raised above (what is the concrete statement of the current theorems, how do you define taking of gradients, and how does your efficient IF compare with the efficient IF of Koh and Liang 2017), I had two other questions:

1. Lemma 4.2 is listed as being from Danskin 2012, but skimming that paper I could not find this lemma. Do you have a more precise theorem you can point to in this paper / could you try to give some more intuition of their statement and proof (especially of the second part that says that the gradient of loss at $\omega^\star$ should be $0$)?

2. Lines 227-233: tries to give some intuition on the difference between IF and HIF, but I couldn't follow it. Even in the traditional ERM-setting, IF neglects certain changes to the Hessian (due to sample drops / dependence on model parameters). Can you give some intuition on why this effect is different? Is it because it somehow affects the leading term when viewing HIF as a linear approximation of sample reweighting?

---

### Official Review · Reviewer_pCw6 · 2025-11-03

**Soundness:** 2
**Presentation:** 2
**Contribution:** 1
**Rating:** 2
**Confidence:** 4

**Summary:**

- This paper proposes data valuation methods for SAM, utilizing the influence function.
- The authors suggest two data evaluation methods, Hessian-based and gradient trajectory-based methodology, where each has its merits and demerits.
- The proposed method enables revealing helpful and harmful training data instances, which leads to better performance.

**Strengths:**

- The paper is well-structured and clearly written.
- The paper is theoretically grounded.

**Weaknesses:**

- The experiment is limited to the image classification task. It would be more convincing if the authors supplemented the experiments with other tasks as well.
- Thorough comparison against baselines is required both for the experiments and computational complexity.

**Questions:**

- Could you provide the complexity analysis against baselines?
- In the qualitative analysis for the most helpful vs harmful training example, some images do not align with the human evaluation of the helpful/harmful valuation, at least for the MNIST case. Why do you think this phenomenon happens? Also, what is the classification accuracy for the harmful data instances?
- Why are the baselines missing in Table 1, Figure 1, etc.?

---

### Note · Authors · 2025-11-13

I have read and agree with the venue's withdrawal policy on behalf of myself and my co-authors.